# Srsf10 and the minor spliceosome control tissue-specific and dynamic SR protein expression

Stefan Meinke[1], Gesine Goldammer[1], A Ioana Weber[1,2], Victor Tarabykin[2], Alexander Neumann[1†], Marco Preussner[1]*, Florian Heyd[1]*

[1]Freie Universität Berlin, Institute of Chemistry and Biochemistry, Laboratory of RNA Biochemistry, Berlin, Germany; [2]Institute of Cell Biology and Neurobiology, Charité-Universitätsmedizin Berlin, corporate member of Freie Universität Berlin, Humboldt-Universität zu Berlin, and Berlin Institute of Health, Berlin, Germany

**Abstract** Minor and major spliceosomes control splicing of distinct intron types and are thought to act largely independent of one another. SR proteins are essential splicing regulators mostly connected to the major spliceosome. Here, we show that *Srsf10* expression is controlled through an autoregulated minor intron, tightly correlating Srsf10 with minor spliceosome abundance across different tissues and differentiation stages in mammals. Surprisingly, all other SR proteins also correlate with the minor spliceosome and *Srsf10*, and abolishing *Srsf10* autoregulation by Crispr/Cas9-mediated deletion of the autoregulatory exon induces expression of all SR proteins in a human cell line. Our data thus reveal extensive crosstalk and a global impact of the minor spliceosome on major intron splicing.

**\*For correspondence:**
mpreussner@zedat.fu-berlin.de (MP);
florian.heyd@fu-berlin.de (FH)

**Present address:** †Omiqa Corporation, c/o Freie Universität Berlin, Altensteinstraße, Germany

**Competing interests:** The authors declare that no competing interests exist.

## Introduction

Alternative splicing (AS) is a major mechanism that controls gene expression (GE) and expands the proteome diversity generated from a limited number of primary transcripts (*Nilsen and Graveley, 2010*). Splicing is carried out by a multi-megadalton molecular machinery called the spliceosome of which two distinct complexes exist. The more abundant major spliceosome that consists of the U1, U2, U4, U5, U6 small nuclear ribonucleoprotein particles (snRNPs) and multiple non-snRNP splicing factors. Additionally, the cells of most eukaryotes contain the minor spliceosome, which is composed of the minor-specific snRNPs U11, U12, U4atac, U6atac, and the shared U5 snRNP. While the major spliceosome catalyzes splicing of around 99.5% of all introns, mainly so-called U2-type introns with the characteristic GT-AG splice sites, the minor spliceosome recognizes introns of the U12-type containing non-consensus AT-AC splice sites and distinct branch point and polypyrimidine sequences (*Jackson, 1991*; *Turunen et al., 2013*). However, despite its low abundance, the minor spliceosome plays a fundamental role in ontogenesis, as deficiencies in minor spliceosome activity or minor intron splicing are lethal or result in developmental defects and disorders (*Doggett et al., 2018*; *Verma et al., 2018*).

AS is highly regulated by *cis*-acting splicing enhancer and silencer elements, which are recognized by various RNA binding proteins, such as SR proteins and heterogeneous nuclear ribonucleoproteins (*Hastings et al., 2001*). The protein family of SR proteins, with 13 canonical members in humans, is characterized by an arginine and serine rich domain (RS domain) (*Manley and Krainer, 2010*). Aside from their role in mRNA nuclear export and GE they are essential regulators of AS (*Long and Caceres, 2009*). Every SR protein contains ultraconserved elements in alternative exons that control the presence of premature translation termination codons (PTC). This allows them to regulate their own abundance through nonsense-mediated decay (NMD) (*Lareau et al., 2007*). While many SR

proteins and RBPs use autoregulation to maintain a stable expression level (*Müller-McNicoll et al., 2019*), their expression level changes in a tissue-specific manner (*Wang et al., 2008*; *Olthof et al., 2019*). Therefore, mechanisms aside from autoregulation are most likely employed to control SR protein levels under different conditions, for instance in different tissues or during differentiation. However, the mechanisms that coordinately regulate SR protein expression levels remain elusive.

SRSF10 is a unique SR protein, as it activates splicing in its phosphorylated state but becomes a general splicing inhibitor upon dephosphorylation (*Feng et al., 2008*; *Zhou et al., 2014*). We used SRSF10 as a case study of how tissue-specific differences in SR protein levels can be achieved by employing an autoregulatory feedback loop. We show that Srsf10 recognizes a highly conserved splicing enhancer element within its own pre-mRNA, which results in the production of a non-protein coding mRNA isoform and thereby the regulation of its own expression level. An additional layer of *Srsf10* regulation is added by the presence of competing major and minor splice sites which control this autoregulatory AS event. The minor splice site leads to the formation of the protein-coding mRNA, whereas splicing mediated by the major spliceosome leads to the non-protein-coding mRNA. Consequently, *Srsf10* levels correlate with the level of the minor spliceosome in a tissue- and developmental stage-specific manner. Surprisingly, we also found that the expression levels of most other SR proteins correlate with *Srsf10* expression. This is directly mediated through the levels of *Srsf10* and the competition between major and minor splice sites, as CRISPR/Cas9-mediated removal of the autoregulatory exon 3 of *Srsf10* increases not only the expression of *Srsf10*, but also the expression of the other SR proteins. These data connect the minor spliceosome with *Srsf10* and SR protein expression in a tissue- and differentiation state-specific manner. We thus reveal a mechanism that coordinately controls SR protein expression in different cellular conditions and that connects the minor spliceosome with global (alternative) splicing of major introns.

## Results and discussion

### Srsf10 autoregulates its own splicing and expression

Autoregulation has been described for many RBPs including most SR proteins (*Lejeune et al., 2001*; *Sureau, 2001*), but not for Srsf10. *Srsf10* represents a particularly interesting example as its conserved region contains two competing 5' splice sites in exon 2 (E2), which are recognized by either the minor or the major spliceosome. The upstream (up) minor splice site is coupled to E4 inclusion and production of a protein coding mRNA, while use of the downstream (dn) major splice site is coupled to E3 inclusion, the presence of a PTC and the use of an alternative polyadenylation site in E3 (*Figure 1A*). The dn-E3 variant is not a canonical NMD target, as the stop codon in E2 is less than 50 nucleotides upstream of the E2/3 junction (*Nagy and Maquat, 1998*) and could thus encode for a hypothetical short protein (Srsf10-s, see below). To investigate whether AS of the competing minor and major splice site in exon 2 of *Srsf10* depends on an autoregulatory feedback loop, we generated an *Srsf10* minigene containing mouse exons 2 to 4 (*Figure 1B*, top). We transfected this minigene into human HeLa cells and investigated AS after knocking down the endogenous *SRSF10*. These experiments revealed strong autoregulation, as *SRSF10* knockdown decreased the dn-E3 isoform and increased the up-E4 product and retention of intron 2 (IR, *Figure 1B*, bottom). We confirmed the knockdown of *SRSF10* (*Figure 1—figure supplement 1A and B*) and observed a reduced E3/E4 ratio for endogenous *SRSF10* mRNA (*Figure 1—figure supplement 1B*), consistent with SRSF10 activating E3 inclusion.

AS of *Srsf10* results in three possible protein isoforms: SRSF10-fl (inclusion of exon 7a), SRSF10-2 (inclusion of exon 7b, see *Figure 1A*), and a hypothetical protein resulting from exon 3 inclusion (stop codon within the dn-E2 sequence, see *Figure 1A*). To investigate the activity of these protein isoforms in regulating AS, we performed rescue experiments with GFP-tagged Srsf10 mouse variants (not targeted by the human-specific siRNA). A Western blot against Srsf10 shows expression of GFP-Srsf10-fl and GFP-Srsf10-2 close to endogenous levels, while GFP-Srsf10-s was not detectable (*Figure 1—figure supplement 1C*, top). Consistently, Srsf10-fl and Srsf10-2 clearly rescue exon 3 missplicing caused by the knockdown of endogenous *SRSF10*, while transfecting Srsf10-s has no effect on *Srsf10* AS (*Figure 1C*). We obtained the same result when overexpressing the different Srsf10 variants with no knockdown of the endogenous protein, confirming Srsf10-fl and Srsf10-2 as activators of exon 3 inclusion and Srsf10-s as a barely expressed protein (*Figure 1—figure supplement 1C*).

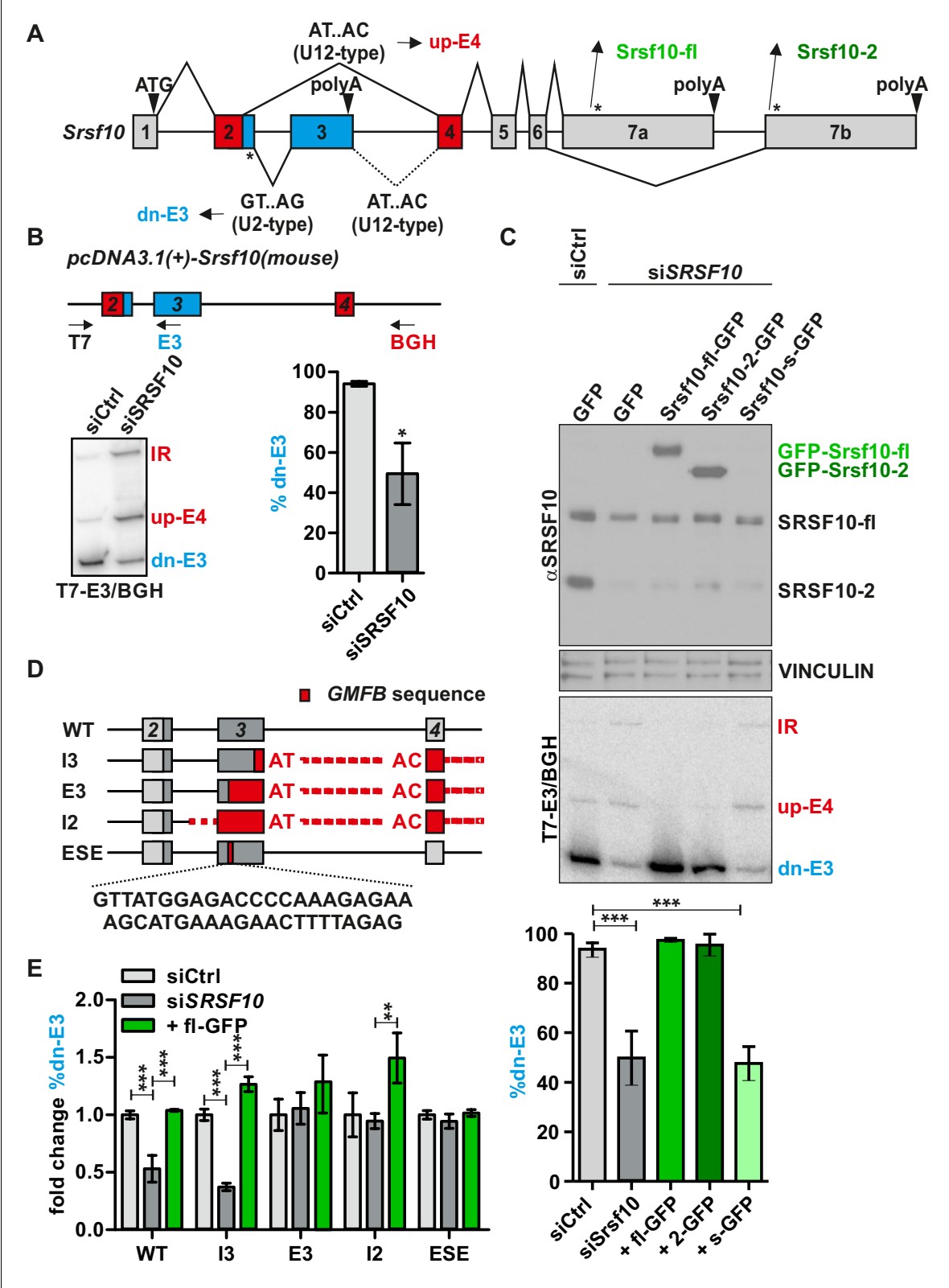

**Figure 1.** Srsf10 autoregulates its own splicing through a conserved enhancer in exon 3. (**A**) Schematic of the exon/intron structure of Srsf10. Usage of the downstream (dn) major splice site in Srsf10 exon 2 (GT.AG, U2-type) leads to exon 3 inclusion and a non-protein coding isoform, while usage of the upstream minor splice site (AT.AC, U12-type) results in exon 4 inclusion. A minor 5' splice site in exon 3 is present but not used in the endogenous context (dotted lines). * indicate stop codons. (**B**) Srsf10 minigene splicing upon siRNA-mediated knockdown of endogenous Srsf10. Top: exon/intron

*Figure 1 continued on next page*

Figure 1 continued

structure of the Srsf10 minigene reporter containing mouse exons 2 to 4 (and complete intervening introns) with indicated primer binding sites (arrows). HeLa cells were transfected with control siRNA (siCtrl) or against human Srsf10 (siSrsf10), incubated for 24 hr followed by minigene transfection. After 48 hr splicing was analyzed with the indicated primers. Bottom: exemplary gel and quantification of the dn-E3 isoform (n = 5, mean ± SD). (C) Knockdown and rescue of SRSF10. Top: Western Blot of SRSF10 after siRNA-mediated knockdown and transfection with overexpression vectors for the different GFP-tagged Srsf10 isoforms. VINCULIN was used as loading control. Middle: Exemplary gel of Srsf10 minigene splicing upon knockdown and rescue. Bottom: Quantification of the dn-E3 isoform (n ≥ 3, mean ± SD). (D) Exon/intron structure of the Srsf10 minigenes used for mutational analysis. Exon and intron sequences were replaced by sequences containing a minor intron from glia maturation factor beta (Gmfb exons 4 to 5 including the minor intron 4, marked in red). Below the sequence of the identified ESE is shown. (E) Quantification of Srsf10 minigene splicing upon knockdown and rescue. HeLa cells were transfected with the mutated minigenes (D) and analyzed as in (B). Splicing of mutants is shown relative to the wt from (B) and for each mutant relative to the Ctrl siRNA (n = 5, mean ± SD). Student's t test-derived p values *p<0.05, **p<0.01, ***p<0.001.

The online version of this article includes the following figure supplement(s) for figure 1:

**Figure supplement 1.** Srsf10 autoregulates its own splicing.
**Figure supplement 2.** Srsf10 autoregulates its own splicing through a conserved enhancer in exon 3.
**Figure supplement 3.** Alignment of the 5' sequence of Srsf10 exon 3, which includes the identified ESE region (marked in red), showing high conservation across 7 mammals.

Since the presence of Srsf10-s is hardly detectable, even with a stabilizing GFP tag, we assume that this protein variant is highly instable and does not have a biological function. To compare the activities of Srsf10-fl and Srsf10-2, we performed titration experiments. This demonstrated highly sensitive *Srsf10* autoregulation, with Srsf10-fl, despite its lower expression levels, being a more potent activator of E3 inclusion than Srsf10-2 (*Figure 1—figure supplement 1D*). These data identify Srsf10 as an activator of the major intron between exons 2 and 3 and suggest that Srsf10 autoregulates its own expression level via a negative feedback loop, as higher Srsf10 levels result in the formation of the non-protein-coding dn-E3 isoform.

To identify the cis-regulatory element required for autoregulation, we used systematic mutational analysis of the *Srsf10* minigene (*Figure 1D*, *Figure 1—figure supplement 2, A–D*). First, we replaced sequences downstream of exon 3 by human Gmfb sequences from exons 4 to 5, including a minor intron (mutant I3, see 'Material and Methods' for details). This reflects the endogenous situation, as *Srsf10* exon 3 contains a minor 5' splice site, which, however, is rarely used, since the polyadenylation site in exon 3 appears to be dominant (*Figure 1A*). The resulting minigene clearly remains responsive to *SRSF10* knockdown and overexpression (*Figure 1E*, I3). In contrast, replacing sequences starting from exon 3 (E3) or in intron 2 (I2) by *Gmfb* sequences results in splicing unresponsive to *SRSF10* knockdown and barely responsive to SRSF10 overexpression (*Figure 1E*; mutants E3 and I2). These data suggest that SRSF10 controls its own splicing via binding to exon 3, which, indeed, contains a GA-rich element representing the previously identified SRSF10 consensus binding site (*Shin and Manley, 2002*; *Zhou et al., 2014*). Replacing nucleotides 17 to 60 of exon 3 by *GMFB* exon 4 sequence was sufficient to abolish SRSF10-mediated AS (*Figure 1D and E*, ESE; *Figure 1—figure supplement 3*), thus identifying this GA-rich element as an SRSF10-dependent exonic splicing enhancer (ESE). Together, these data identify a highly conserved element in *Srsf10* exon 3 which is necessary for an autoregulatory feedback loop that controls Srsf10 expression levels.

## The minor spliceosome controls Srsf10 expression

Exon 2 of *Srsf10* contains two competing 5' splice sites, which are specifically recognized by either the minor or the major spliceosome. To investigate the relevance of these splice sites for *Srsf10* autoregulation, we generated mutated minigenes containing either only major or only minor splice site (*Figure 2A*) and analyzed AS of the resulting minigenes. Mutated minigenes remained clearly responsive to SRSF10 knockdown and rescue, demonstrating that SRSF10 can regulate AS through both major and minor spliceosomes. However, in the control conditions (siCtrl), we observed that both minor-only or major-only minigenes show a strong increase in the use of exon 4 (*Figure 2B*). Exon 3 is hardly included at all in any of the two minigenes. Additionally, in the presence of a directly competing upstream splice site in exon 2, the downstream splice site is no longer used (*Figure 2B*, *Figure 2—figure supplement 1*). These data indicate that, in vivo, the use of the minor splice site that leads to productive *Srsf10* splicing, is reduced through the presence of a competing major splice site. This arrangement could render *Srsf10* expression susceptible to dynamic control through

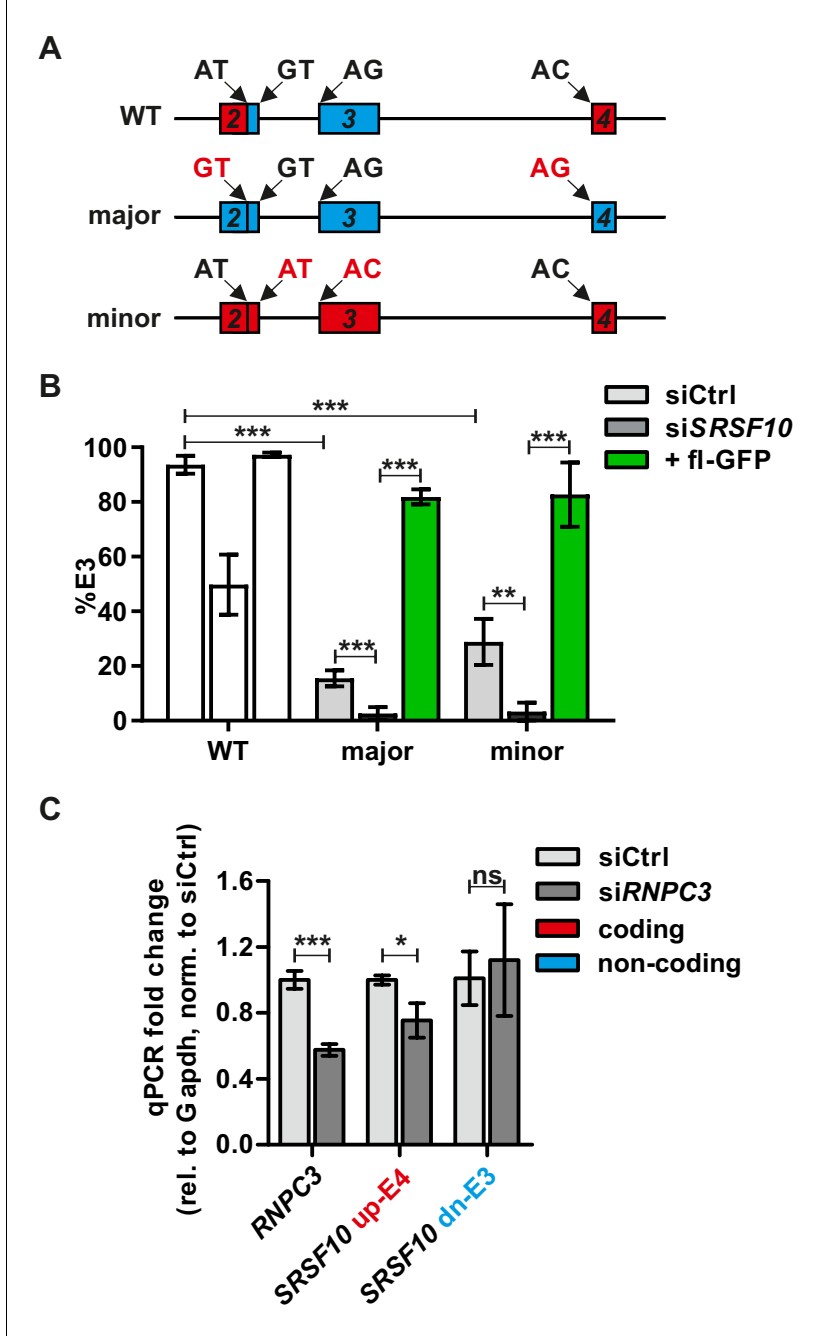

**Figure 2.** The minor spliceosome controls Srsf10 expression. (**A**) Schematic of the exon/intron structure of the Srsf10 WT and mutated minigenes harboring only major (GT.AG, blue) or minor (AT.AC, red) splice sites. Mutated splice sites are marked in red. (**B**) Minigenes from (**A**) were analyzed as in *Figure 1*. Quantification of splicing-sensitive RT-PCRs is shown relative to the WT from *Figure 1B* (n = 4, mean +/- SD). A representative gel is shown in *Figure 2—figure supplement 1*. (**C**) RT-qPCRs confirm siRNA-mediated knockdown of Rnpc3 (left) and changes in Srsf10 expression levels in HEK293. Expression relative to Gapdh and normalized to siCtrl (n = 3, mean +/- SD). Student's t test-derived p values *p<0.05, **p<0.01, ***p<0.001.

The online version of this article includes the following figure supplement(s) for figure 2:

**Figure supplement 1.** Competition of minor and major splice sites favors the downstream 5' splice site A representative gel of only minor or major splice site containing minigenes.

alterations in the activity of the minor spliceosome. To directly investigate this hypothesis, we inhibited minor spliceosome activity by performing an siRNA-mediated knockdown of the essential U11/U12 snRNP component *RNPC3* (*Figure 2C*). Indeed, the expression of the coding *SRSF10* mRNA (up-E4) was significantly decreased, while the levels of the non-coding dn-E3 mRNA are unaffected or slightly increased (*Figure 2C*). This result indicates that the abundance of the minor spliceosome directly correlates with *Srsf10* GE through controlling productive vs. non-productive AS. Furthermore, regulation through the activity of the minor spliceosome appears to, at least partially, overrule the autoregulatory feedback loop. This suggests a model in which the activity of the minor spliceosome sets the expression level of *Srsf10*, which is then maintained through autoregulation.

## Minor spliceosome and SR protein expression correlate in a tissue-specific manner

To investigate the relevance of this mechanism in vivo, we analyzed the correlation of *Srsf10* GE levels with expression of the minor spliceosome component *Rnpc3* across 25 different mouse tissues (*Figure 3A*). Calculated transcripts per million (tpm) values using Whippet Quant (*Sterne-Weiler et al., 2018*) revealed clear tissue-specific expression patterns for both *Srsf10* and *Rnpc3*. Both genes show the lowest GE in blood cells and the highest in thymus (*Figure 3A*). Notably, a linear regression fit revealed an almost perfect correlation of *Srsf10* and *Rnpc3* expression across the 25 investigated tissues ($R^2$=0.85, p<0.0001, *Figure 3B*). Similarly, *SRSF10* and *RNPC3* GE levels correlate across 31 human tissues ($R^2$=0.33, p=0.0006, *Figure 3—figure supplement 1A*). In contrast, the levels of the housekeeping gene *Gapdh*, which contains no minor intron, do not correlate with *Rnpc3* ($R^2$=0.01, p=0.4071, *Figure 3C*). Similar correlation coefficients were obtained with gene expression values determined independently using Salmon (*Patro et al., 2017*; *Figure 3—figure supplement 1B and C*). Globally, minor intron-containing genes correlate much more strongly with *Rnpc3* levels than a randomly chosen group of expression level-matched genes containing only major introns (*Figure 3D*), indicating that *Rnpc3* levels represent an adequate indicator for minor spliceosome activity. Consistently, *Rnpc3* levels, and therefore *Srsf10* levels, also correlate with the expression levels of two other minor spliceosome components, namely *Snrnp25* and *Snrp48* (*Figure 3—figure supplement 1D and E*). These in vivo GE data are consistent with our model that minor spliceosome activity controls *Srsf10* levels. As an additional model system, we compared *Rnpc3* and *Srsf10* levels during neuronal differentiation of mouse embryonic stem cells (ES cells). GE levels correlate significantly ($R^2$=0.34, p=0.0006, *Figure 3—figure supplement 1F*), while *Gapdh* levels do not correlate with *Rnpc3* ($R^2$=0.02, p=0.4365, *Figure 3—figure supplement 1G*). Again, globally, minor intron-containing genes show a stronger correlation with *Rnpc3* levels than genes containing only major introns (*Figure 3—figure supplement 1H*). The *Srsf10*/*Rnpc3* correlation in ES cell differentiation is less pronounced – also with Salmon derived tpm values (*Figure 3—figure supplement 1I*) – indicating other factors influencing gene expression. Normalization of gene expression using DESeq2 (*Love et al., 2014*) strongly increases this correlation (*Figure 3—figure supplement 1J*), which is consistent with a direct role of the minor spliceosome in regulating *Srsf10* levels across different tissues and development stages. In summary, together with our minigene and knockdown results, these data indicate that the activity of the minor spliceosome controls productive vs unproductive *Srsf10* splicing and expression levels during development and in a tissue-specific manner.

## SRSF10 and the minor spliceosome control tissue-specific and dynamic SR protein expression

To investigate whether the minor spliceosome exclusively controls *Srsf10* expression, we next examined the expression levels of all other SR proteins in a tissue- and developmental stage-specific manner. Surprisingly, we found that the expression of all SR proteins correlates with the minor spliceosome, represented by *Rnpc3*, in the 25 investigated tissues ($R^2$>0.42, p<0.0001, *Figure 4A* and *Figure 4—figure supplement 1A*). Additionally, we observed a highly similar expression pattern for all SR proteins during neuronal differentiation of mouse ES cells (*Figure 4B* and *Figure 4—figure supplement 1B*) and a significant correlation with *Rnpc3* (*Figure 4—figure supplement 1C*). To experimentally confirm these results in vivo, we isolated RNA from mouse cerebral cortices from embryonic days (E) 12.5 and 15.5. RT-qPCR *analysis* revealed a significant increase of *Rnpc3* GE from E 12.5 to E 15.5 and, in parallel, Srsf10 levels also increased (*Figure 4C and D*). This is

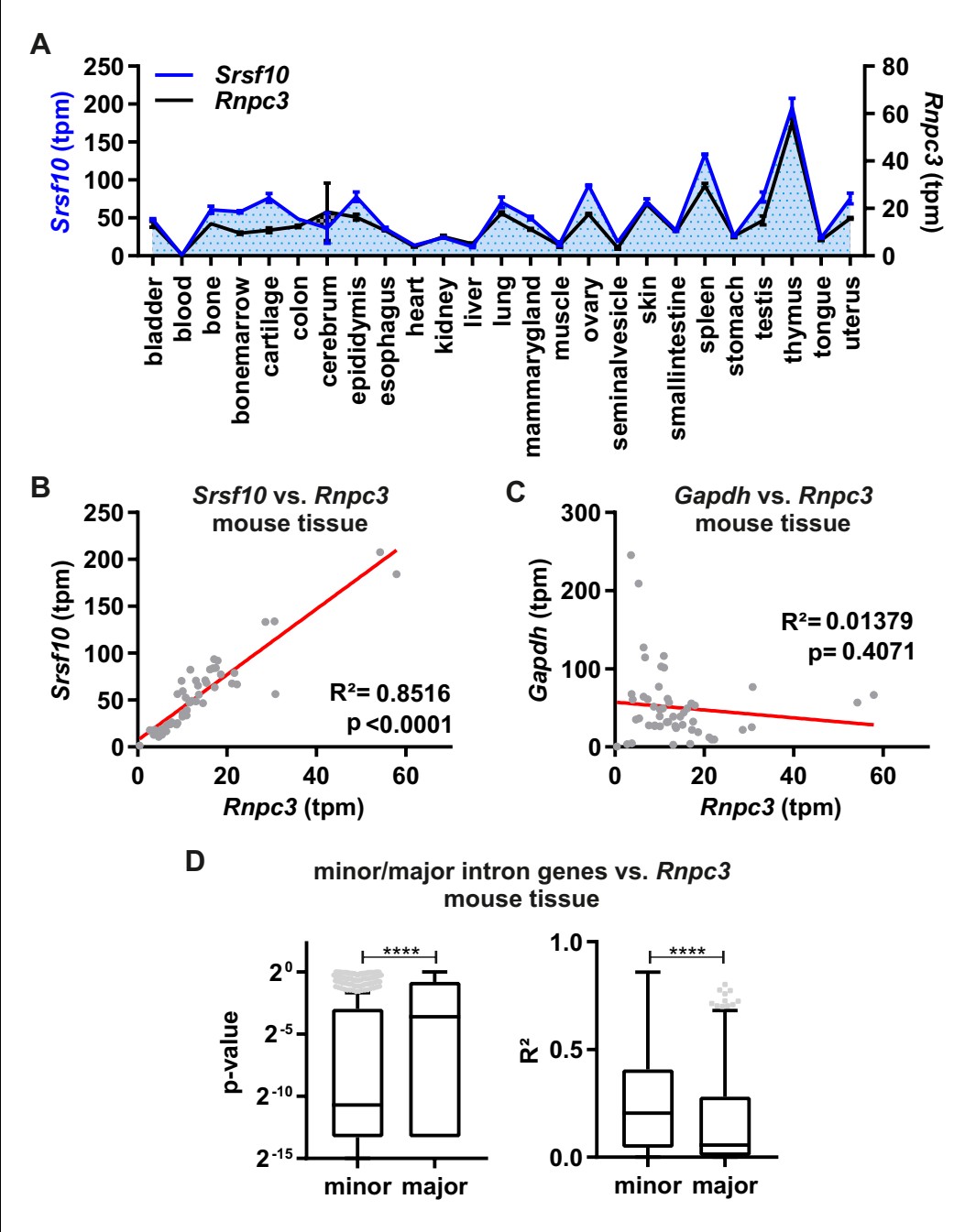

**Figure 3.** Minor spliceosome and Srsf10 expression correlate in a tissue-specific manner. (**A**) Relative GE levels of Srsf10 and Rnpc3 across 25 mouse tissues (x-axis). Transcripts per million (tpm) values were calculated using Whippet (***Sterne-Weiler et al., 2018***) (n ≥ 2, mean +/- SEM). (**B, C**) Linear regression fit for comparison of Srsf10 (**B**) and Gapdh (**C**) GE/tpm values with Rnpc3 across the 25 different mouse tissues. Goodness of fit is represented by $R^2$ and p-values. (**D**) Calculated p-values (left) and $R^2$ values (right) of a global correlation analysis of Rnpc3 with minor intron containing genes (n = 587) or randomly chosen expression matched genes, containing only major introns (n = 629). Statistical significance was determined by an unpaired t-test ****$p<0.0001$.

The online version of this article includes the following figure supplement(s) for figure 3:

**Figure supplement 1.** Minor spliceosome and Srsf10 expression correlate in a tissue-specific manner and during development.

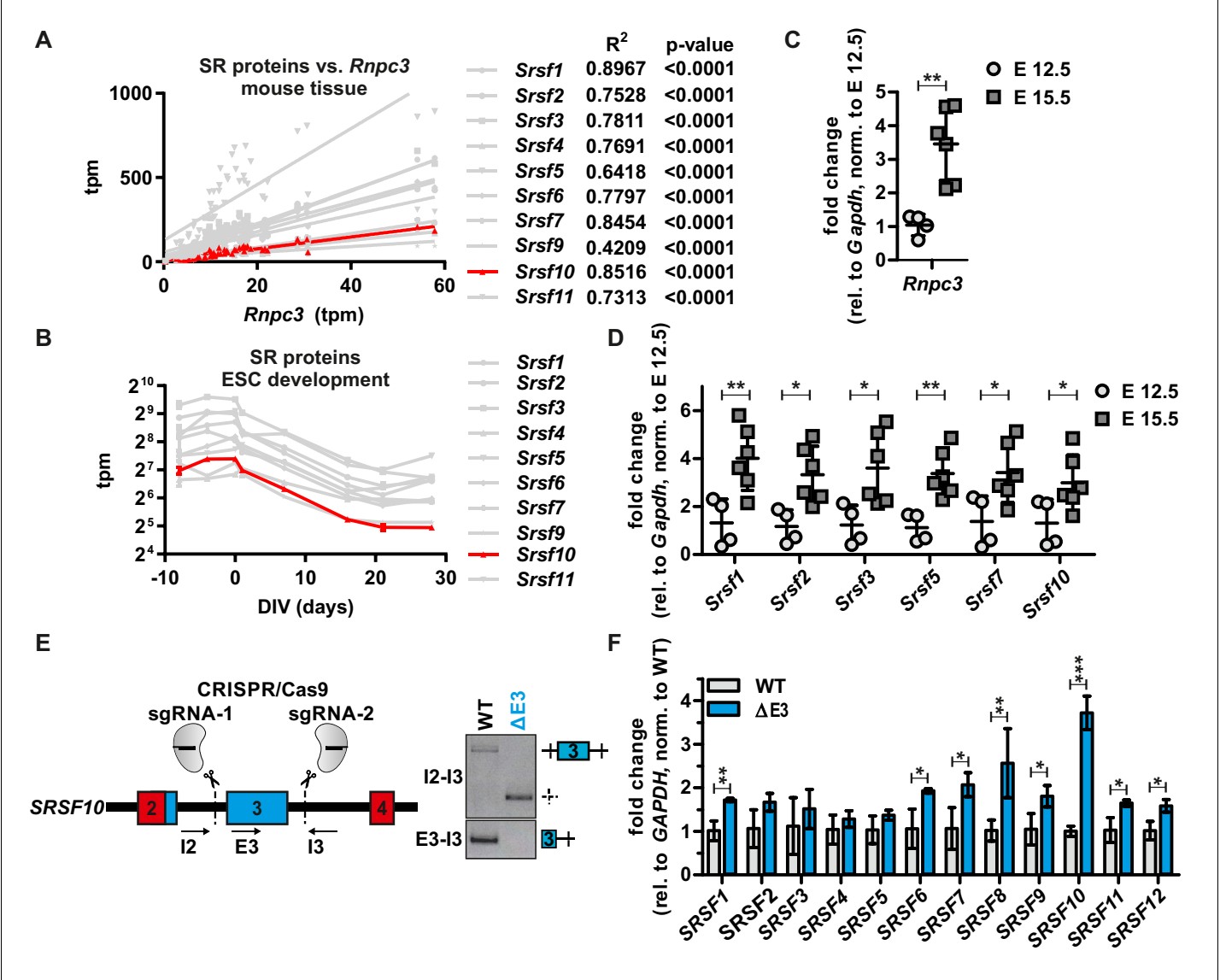

**Figure 4.** SRSF10 and the minor spliceosome control tissue-specific and dynamic SR protein expression. (A) Linear regression analysis of calculated tpm values for SR proteins against Rnpc3 in 25 different mouse tissues. $R^2$ and p-values are shown on the right. See also *Figure 4—figure supplement 1A* with all SR proteins colored. Due to generally low expression levels Srsf12 was omitted. (B) Comparison of tpm values of SR proteins during mouse ES cell differentiation ($n \geq 3$, mean ± SD). See also *Figure 4—figure supplement 1B and C*. (C) Rnpc3 GE levels in mouse cortex samples of the developmental stages E 12.5 and E 15.5 (relative to Gapdh, normalized to time point E 12.5) ($n \geq 4$, mean ± SD). (D) SR protein expression in mouse cortex samples of the developmental stages E 12.5 and E 15.5. For Srsf10 only the functional up-E4 isoform was analyzed (relative to Gapdh, normalized to time point E 12.5) ($n \geq 4$, mean ± SD). (E) Generation of clonal HEK293 cell lines lacking the regulatory Srsf10 exon 3. Left: Schematic of CRISPR/Cas9-mediated deletion of Srsf10 exon 3. Arrows indicate primer binding sites. Right: genotyping PCR on genomic DNA. (F) Relative SR protein expression in WT HEK293 cells or Srsf10ΔE3 cells. For Srsf10 only the functional up-E4 isoform was analyzed (relative to Gapdh, normalized to WT) ($n = 3$, mean +/- SD). Statistical significance was determined by an unpaired t-test *$p<0.05$, **$p<0.01$, ***$p<0.001$.
The online version of this article includes the following figure supplement(s) for figure 4:

**Figure supplement 1.** SRSF10 and the minor spliceosome control tissue specific and dynamic SR protein expression.
**Figure supplement 2.** Model for minor spliceosome-mediated control of SR proteins AS of SRSF10 mirrors minor spliceosome activity.

consistent with our model that higher minor spliceosome activity (indicated by higher *Rnpc3* abundance) results in preferential usage of the *Srsf10* up-E4 minor splice site, which leads to an increase in protein coding *Srsf10* mRNA. Consistent with SR protein expression patterns from different tissues or stem cell development in RNA-Seq data (*Figure 4A and B*), all other tested SR proteins are also

upregulated during the transition from E 12.5 to E 15.5 (*Figure 4D*), which, again, indicates a co-regulation of SR proteins, even though *Srsf10* is the only SR protein that contains a minor intron. As SR proteins are known to cross-regulate each other (*Bradley et al., 2015*), we hypothesized that a change in *Srsf10* levels could directly influence the levels of other SR proteins. To test this, we generated a CRISPR/Cas9-edited cell line lacking the non-productive *SRSF10* exon 3 (*Figure 4E*, left). Homozygous removal of exon 3 was confirmed by PCRs at the genomic level (*Figure 4E*, right) and, as expected, we observe increased *SRSF10* expression (*Figure 4F*, see *Figure 4—figure supplement 1D* for protein expression). Interestingly, we observe a stronger increase in the less active SRSF10-2 isoform (*Figure 4—figure supplement 1D*), indicating that AS of the last exon could be used to partially compensate for the loss of autoregulation via exon 3. Next, to examine whether manipulation of *SRSF10* autoregulation is sufficient to change the overall activity of SRSF10, we analyzed AS of SRSF10 target exons (*Zhou et al., 2014*; *Wei et al., 2015*). Consistent with increased *SRSF10* expression, the inclusion of alternative exons in *BCLAF1*, *PTBP2* and *ZFP207* is promoted in the CRISPR/Cas9-edited cells, and the opposite is observed by knockdown of *SRSF10* (*Figure 4—figure supplement 1E*). Notably, increased SRSF10 in our CRISPR/Cas9-edited cell line was sufficient for subtle upregulation of the mRNA levels of all other SR proteins (*Figure 4F*). In addition, reduced *SRSF10* expression upon *RNPC3* knockdown also correlates with slightly decreased expression of most other SR proteins (*Figure 4—figure supplement 1F*). In summary, these data suggest a model where minor spliceosome activity directly controls *SRSF10* levels, which is sufficient to change expression levels of the other SR proteins in a tissue- and differentiation state-specific manner (*Figure 4—figure supplement 2*). An exciting question that remains is how SRSF10 is able to coordinately regulate the abundance of the other SR proteins. Cross-regulatory mechanisms are known for many RBPs (*Kumar and Lopez, 2005*; *Rossbach et al., 2009*) and we therefore speculate that SRSF10 could regulate other SR proteins (and potentially other RBPs) by repressing the formation of NMD-targeted isoforms from their pre-mRNAs. This could be mediated either directly, by binding to the respective pre-mRNAs, or indirectly, through interactions with other SR proteins. Additionally, differences in SR protein abundance could be achieved by changing their activity. Higher mRNA abundance could be the consequence of reduced SR protein activity, resulting in reduced autoregulatory NMD exon inclusion and therefore higher mRNA expression levels (*Ni et al., 2007*). The activity of SR proteins is, amongst others, controlled by their phosphorylation level (*Goldammer et al., 2018*), and could be controlled by a direct or indirect effect of SRSF10 on the regulating kinases and phosphatases. While these mechanistic details remain to be investigated, the control of SR protein levels through the minor spliceosome and SRSF10 is of fundamental importance, as SR protein levels will have consequences for the splicing of most major introns. Our data thus reveal a mechanism through which the activity of the minor spliceosome controls major intron splicing in a tissue- and differentiation state-specific manner. This may also be relevant for diseases caused by minor spliceosome deficiencies (*Jutzi et al., 2018*; *Verma et al., 2018*), as misregulation of SR proteins and consequently, defects in major intron splicing (*Cologne et al., 2019*), may contribute to the observed phenotypes.

# Materials and methods

**Key resources table**

| Reagent type (species) or resource | Designation | Source or reference | Identifiers | Additional information |
|---|---|---|---|---|
| Gene (*Mus musculus*) | Srsf10 | ncbi | GeneID:14105 | Exons 2–4 |
| Gene (*Homo sapiens*) | Gmfb | ncbi | GeneID:2764 | Exon 4–5 |
| Gene (*Homo sapiens*) | Srsf10 | ncbi | GeneID:10772 | |
| Cell line (*Homo sapiens*) | HeLa | ATCC | RRID:CVCL_0030 | |

*Continued on next page*

*Continued*

| Reagent type (species) or resource | Designation | Source or reference | Identifiers | Additional information |
|---|---|---|---|---|
| Cell line (*Homo sapiens*) | HEK293 | ATCC | RRID:CVCL_0045 | |
| Cell line (*Homo sapiens*) | HEK293ΔE3 | This paper | | CRISPR/Cas9-mediated deletion of Srsf10 exon 3; see *Figure 4* and Materials and methods Part |
| Biological sample (*Mus musculus*, NMRI strain) | Cortices | This paper | | developmental stages E 12.5 and E 15.5; see *Figure 4* and Materials and methods Part |
| Antibody | anti-FUSIP1 (T-18) (human, monoclonal) | Santa Cruz Biotechnology | RRID:AB_1123037 | 1:1000 |
| Antibody | Anti-GFP (B-2) (monoclonal) | Santa Cruz Biotechnology | RRID:AB_627695 | 1:5000 |
| Antibody | anti-Vinculin (H-300) (rabbit, polyclonal) | Santa Cruz Biotechnology | RRID:AB_2214507 | 1:1000 |
| Antibody | anti-hnRNP L (4D11) (human, monoclonal) | Santa Cruz Biotechnology | RRID:AB_627736 | 1:10000 |
| Recombinant DNA reagent | Mouse Srsf10 minigene | This paper | | See *Figure 1* + with *Figure 1—figure supplements 1* and *2*; and Materials and methods part |
| Recombinant DNA reagent | mouse Srsf10-fl-GFP | This paper | | See *Figure 1* + with *Figure 1—figure supplements 1* and *2*; and Materials and methods part |
| Recombinant DNA reagent | mouse Srsf10-2-GFP | This paper | | See *Figure 1* + with *Figure 1—figure supplements 1* and *2*; and Materials and methods part |
| Recombinant DNA reagent | mouse Srsf10-s-GFP | This paper | | See *Figure 1* + with *Figure 1—figure supplements 1* and *2*; and Materials and methods part |
| Recombinant DNA reagent | PX459 vector | Kindly provided by Stefan Mundlos | RRID:Addgene_62988 | For sgRNA cloning and transfection |
| Recombinant DNA reagent | pEGFP-N3 | Clontech | | SRSF10 expression construct |
| Recombinant DNA reagent | pcDNA3.1(+) | Invitrogen | Catalog no: V79020 | Minigene cloning |
| Sequence-based reagent | siRNA against human Srsf10 (siSrsf10) | This paper | GCGUGAAUUUGGUUAUdTdT | Knockdown of endogenous Srsf10 |
| Sequence-based reagent | siRNA against human Rnpc3 (siRnpc3) | This paper | GAAAGAAGGUCGUAUGAAAdTdT | Knockdown of endogenous Rnpc3 |
| Sequence-based reagent | Control siRNA (siCtrl) | This paper | UUUGUAAUCGUCGAUACCCdTdT | |
| Sequence-based reagent | sgRNA: SRSF10_E3_3fw | Benchling Tool | RRID:SCR_013955 | Sequence: caccgctactttact cggtaagcca; CRISPR/Cas9-mediated deletion of Srsf10 exon 3 |

*Continued on next page*

*Continued*

| Reagent type (species) or resource | Designation | Source or reference | Identifiers | Additional information |
|---|---|---|---|---|
| Sequence-based reagent | sgRNA: SRSF10_E3_3rv | Benchling Tool | RRID:SCR_013955 | Sequence: aaactggcttaccg agtaaagtagc; CRISPR/ Cas9-mediated deletion of Srsf10 exon 3 |
| Sequence-based reagent | sgRNA: SRSF10_E3_5fw | Benchling Tool | RRID:SCR_013955 | Sequence: caccgtgagtttc agaagcatgaat; CRISPR/ Cas9-mediated deletion of Srsf10 exon 3 |
| Sequence-based reagent | sgRNA: SRSF10_E3_5rv | Benchling Tool | RRID:SCR_013955 | sequence: aaacattcatgctt ctgaaactcac; CRISPR/ Cas9-mediated deletion of Srsf10 exon 3 |
| Commercial assay or kit | PowerUp SYBR Green Mastermix | ThermoFisher Scientific | A25742 | RT-qPCR |
| Chemical compound, drug | Roti-Fect | Carl Roth | Order no:P001.1 | Plasmid vector transfection |
| Software, algorithm | GraphPad Prism 7.05 | GraphPad | RRID:SCR_002798 | Statistical analysis |
| Software, algorithm | ImageQuant TL | GE Life Sciences | RRID:SCR_014246 | quantification |
| Software, algorithm | Whippet v0.11 | *Sterne-Weiler et al., 2018* | RRID:SCR_018349 | Tpm calculation |
| Software, algorithm | Salmon v1.2.0 | *Patro et al., 2017* | RRID:SCR_017036 | Tpm calculation |
| Software, algorithm | TxImport v1.14.0 | *Soneson et al., 2015* | RRID:SCR_016752 | Import of transcript counts to R for normalization wit DESeq |
| Software, algorithm | DESeq2 v1.26.0 | *Love et al., 2014* | RRID:SCR_015687 | Transcript counts normalization |

## Tissue cell culture

HEK293 and HeLa cells were cultured in standard conditions. Transfections were done with Rotifect according to the manufacturer's instructions. For siRNA sequences see *Supplementary file 1*. Minigenes were transfected 24 hr after knockdown and RNAs were isolated 48 hr later. For overexpression and rescue experiments, cells were transfected using 0.4 µg of minigenes and 0.4 µg expression vectors for GFP-tagged Srsf10-fl, Srsf10-2 or Srsf10-s (or GFP alone). 48 hr after transfection cells were harvested for protein and/or RNA preparation. We perform monthly test for mycoplasms using PCR. Cell have been tested negative in all tests during the experiments performed for the present study. The cells morphologically appear as expected for Hek293 and HeLa cells respectively. We have used these Hek293 cells in several RNA-Seq experiments and compared gene expression with published Hek293 datasets and found very good correlation.

## Preparation of embryonic mouse cortices for RNA extraction

Colonies of wild type mice of the NMRI strain were maintained in the animal facilities of Charité-Universitätsmedizin Berlin. Tissue collection was performed in compliance with German Animal Welfare Law and regulations imposed by the State Office for Health and Social Affairs Council in Berlin/Landesamt für Gesundheit und Soziales (LAGeSo).

Mice were bred for timed pregnancies, and the date of vaginal plug detection was considered embryonic day 0.5. Pregnancies were timed accordingly, and embryos prepared at the indicated embryonic days. Prior to the preparation of embryos, all tools were thoroughly cleaned with RNAse AWAY solution (Thermo Fisher, cat. no. 10328011). The uteri were dissected into DEPC-treated PBS,

and the embryonic brains quickly transferred to a solution of 2M NaCl in PBS for RNA stabilization. Cortices were dissected and then snap frozen in liquid nitrogen.

## Molecular cloning

For cloning of the different Srsf10 overexpression constructs inserts were amplified from mouse cDNA and cloned into pEGFP-N3 (Clontech) using XhoI and BamHI restriction sites introduced through PCR primers. For minigene cloning, mouse genomic DNA was used as template to amplify *Srsf10* exons 2 to 4 and the product was cloned into pcDNA3.1(+). See *Supplementary file 1* for cloning primer sequences. For mutational analysis *Srsf10* exons/introns were replaced by sequences from the human *GMFB* gene: exons 4 to 5, intron 4 is a minor intron. New inserts were amplified by 1-step or 2-step PCR and cloned into pcDNA3.1(+) using BamHI and XhoI or into the WT minigene using internal restrictions sites (BsrGI for exon 3 and XbaI for intron 3). In the I3 mutant 356 nt of Srsf10 exon 3 were maintained, the downstream sequence was replaced by the 5' splice site of *GMFB* exon 4 (TCGatatcc...) and downstream sequence. For the E3 mutant sequences downstream of the BsrGI site (+16 in exon 3) were replaced by *GMFB* exon 4 to 5 sequence starting with (5'-CACCAGA...). For the ESE mutant nucleotids 17 to 60 of exon 3 were replaced by *GMFB* exon 4 sequence. For the minigene containing only minor splice sites we replaced the major splice sites of *Srsf10* intron 2 by minor splice sites of *GMFB* intron 4 (5' splice site: 5'-TCGatatcc; 3' splice site: 5'-ttctttaacttgagaaaaacCTT). In the minigene containing only major splice sites the upstream 5' splice site of exon 2 and the 3' splice site of exon 4 were replaced by major splice sites from *GMFB* intron 5 or 3, respectively (5' splice site: 5'-TTGgtaagt; 3' splice site: 5'-gctttctctgtggtgccagGGC). All constructs were confirmed by sequencing.

## RT-PCR and RT-qPCR

RT-PCRs and RT-qPCRs were performed as previously described (*Preußner et al., 2017*). See *Supplementary file 1* for primer sequences.

## Western blot

Western Blot analyses were done as previously described (*Preußner et al., 2017*). The following antibodies were used: αSRSF10/FUSIP1 (T-18, Santa Cruz), αGFP (B-2, Santa Cruz), αVINCULIN (H-300, Santa Cruz) and αHNRNPL (4D11, Santa Cruz) antibodies. Western blots were quantified using the ImageQuant TL software.

## RNA-Seq analysis

Transcripts per million (tpm) values were extracted from previously published RNA sequencing data using Whippet version 0.11 (*Sterne-Weiler et al., 2018*) and mouse reference genome mm10. Data for mouse tissues were obtained from WT control samples from SRA study DRP003641 (*Tanikawa et al., 2017*), data for neuronal differentiation of mouse ES cells from SRA study SRP017778 (*Hubbard et al., 2012*). Additionally, skin samples from SRP115206 were analyzed. For comparison, transcript abundances were additionally quantified using salmon version 1.2.0 (*Patro et al., 2017*). To obtain normalized expression counts, transcripts were imported to R using TxImport version 1.14.0 (*Soneson et al., 2015*) and count normalization was performed using DESeq2 version 1.26.0 (*Love et al., 2014*). DESeq2 based normalized expression counts dramatically increased the variance of replicate tissue samples (but not of ES cell differentiation), which is why we chose to analyze only the un-normalized values for the tissue comparisons. To investigate GE levels from different human tissues, fpkm values were directly used from *Uhlén et al. (2015)*. Linear regression fits were performed in GraphPad Prism 7.05. A list of minor intron containing genes is based on *Olthof et al. (2019)*. Genes with a comparable distribution of GE levels were randomly chosen from the remaining only major intron containing genes. The GE of each gene was compared to Rnpc3 levels across tissues or development stages. Unexpressed genes were excluded from the analysis. Downstream analyses were performed using standard Python 2 code and GraphPad Prism.

## Generation of CRISPR/Cas9-edited cell lines

For CRISPR/Cas9-mediated deletion of *SRSF10* exon 3 sgRNAs were designed using the Benchling tool (for sequences see *Supplementary file 1*) and cloned into the PX459 vector. Cells were

transfected using Rotifect following the manufacturer's protocol. 48 hr after transfection, transfected cells were selected with 1 µg/ml puromycin and clonal cell lines were isolated by dilution (*Ran et al., 2013*). Genomic DNA from clonally expanded lines was extracted and analyzed by PCR.

## Acknowledgements

The authors wish to thank members of the Heyd lab for constructive comments and the HPC Service of ZEDAT, Freie Universität Berlin, for computing time. This work was funded by DFG grant 278001972 - TRR 186 to FH and DFG grant TA303/8-1 to VT. IW was supported by a PhD fellowship of the Boehringer Ingelheim Fonds and the Charité Promotionsstipendium. MP is funded by a post-doc stipend of the Peter and Traudl Engelhorn Foundation.

## Additional information

### Funding

| Funder | Grant reference number | Author |
| --- | --- | --- |
| Deutsche Forschungsge-meinschaft | 278001972 - TRR 186 | Florian Heyd |
| Deutsche Forschungsge-meinschaft | TA303/8-1 | Victor Tarabykin |
| Boehringer Ingelheim Fonds | PhD fellowship | A Ioana Weber |
| Charité Promotionsstipendium | | A Ioana Weber |
| Peter and Traudl Engelhorn Foundation | Postdoc stipend | Marco Preussner |

The funders had no role in study design, data collection and interpretation, or the decision to submit the work for publication.

### Author contributions

Stefan Meinke, Data curation, Formal analysis, Investigation, Methodology, Writing - original draft, Writing - review and editing; Gesine Goldammer, Investigation, Methodology; A Ioana Weber, Formal analysis, Investigation, Writing - review and editing; Victor Tarabykin, Resources; Alexander Neumann, Formal analysis; Marco Preussner, Conceptualization, Data curation, Formal analysis, Supervision, Investigation, Methodology, Writing - original draft, Writing - review and editing; Florian Heyd, Conceptualization, Formal analysis, Supervision, Funding acquisition, Methodology, Writing - original draft, Writing - review and editing

### Author ORCIDs

Stefan Meinke (ID) https://orcid.org/0000-0001-5083-3383
Marco Preussner (ID) https://orcid.org/0000-0001-5155-0844
Florian Heyd (ID) https://orcid.org/0000-0001-9377-9882

### Ethics

Animal experimentation: Colonies of wild type mice of the NMRI strain were maintained in the animal facilities of Charité-Universitätsmedizin Berlin. Tissue collection was performed in compliance with German Animal Welfare Law and regulations imposed by the State Office for Health and Social Affairs Council in Berlin / Landesamt für Gesundheit und Soziales (LAGeSo) under licence T102/11.

### Decision letter and Author response

Decision letter https://doi.org/10.7554/eLife.56075.sa1
Author response https://doi.org/10.7554/eLife.56075.sa2

# Additional files

## Supplementary files
• Supplementary file 1. Oligonucleotide sequences of primers, siRNAs, and guide RNAs.

• Transparent reporting form

## Data availability
All data are previously published and publicly available.

The following previously published datasets were used:

| Author(s) | Year | Dataset title | Dataset URL | Database and Identifier |
|---|---|---|---|---|
| Human Genome Center, The University of Tokyo | 2017 | p53 mouse multi-tissue transcriptome analysis | https://www.ncbi.nlm.nih.gov/sra/?term=DRP003641 | NCBI Sequence Read Archive, DRP003641 |
| USAMRICD | 2012 | Deep transcriptional profiling of longitudinal changes during neurogenesis and network maturation in vivo | https://www.ncbi.nlm.nih.gov/sra/?term=SRP017778 | NCBI Sequence Read Archive, SRP017778 |

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
