## [Decision Letter]

Thank you for submitting your article "Srsf10 and the minor spliceosome control tissue-specific and dynamic SR protein expression" for consideration by *eLife*. Your article has been reviewed by two peer reviewers, and the evaluation has been overseen by a Reviewing Editor, Timothy Nilsen, and James Manley as the Senior Editor. The reviewers have opted to remain anonymous.

The reviewers have discussed the reviews with one another and the Reviewing Editor has drafted this decision to help you prepare a revised submission.

As you will see, the reviewers were quite positive about the work and reviewer 3 was very enthusiastic. The reviewing editor was also enthusiastic about the manuscript. Nevertheless, the reviewers have raised some concerns regarding the data and its interpretation. Please address these points as thoroughly as possible via revision. It is not necessary to delete the impact of *Srsf10* expression on other SR proteins.

Reviewer #2:

This paper describes an unusual regulatory feedback process in the alternative splicing of the *Srsf10* gene involving competition between major and minor intron splice sites. Several RNA binding proteins self-regulate their protein levels by inclusion of alternative "poison exons" that include premature stop codons. *Srsf10* appears to have a poison exon 3 who's exclusion requires minor spliceosome activity. This part of the paper seems solid. They then show that there is a general correlation of the expression levels of many SR proteins including *Srsf10*. This has been described before and similar correlations can be seen in ribosomal protein genes indicating that they are under coordinate regulation. The authors then remove the poison exon from the *Srsf10* gene and argue that the results support a key role for *Srsf10* and, by extension, the minor spliceosome in controlling the mRNA levels of SR genes in general. The key results shown in Figure 4E do not, to me, support this idea. The authors speculate that alterations in mRNA levels are due to NMD. A direct demonstration would be more convincing. With the data they show, I think they should focus on the *Srsf10* case alone rather than pushing for a more global mechanism.

Reviewer #3:

In this work, Meinke et al. demonstrate that alternative splicing of *Srsf10* is formed via competition between the major and the minor spliceosome using different splice sites. These in turn lead to functional isoforms that skip exon 3 (minor spliceosome) or inclusion of exon 3 (major spliceosome) in a mostly non functional isoform which encodes a PTC and utilizes an alternative 3' end. The authors show that similar to other Sr proteins Srsf10 regulates its own levels via the inclusion of exon 3 and find a short site in that exon that is sufficient to achieve that regulation. They also perform rescue experiments with SRSF10 mouse variants in human HeLa cells, and a titration experiment with the different isoforms detected (Figure 1). KD of *Rnpc3* of the minor spliceosome changes the splicing and expression of *Srsf10* as expected (Figure 2). Expression of *Srsf10* correlates well with expression of *Rnpc3* across diverse mouse tissues, and as expected *Rnpc3* levels correlate much better with expression of intron containing genes than matched expression levels of genes with only major introns (Figure 3). When removing exon3 control of *Srsf10* levels via CRISPR the expression of *Srsf10* increase by ~3.5fold but six other SR genes expression levels rises significantly, and these too correlate well with expression of *Rnpc3* (Figure 4). The overall regulatory model is summarized in Figure 4—figure supplement 2.

Overall, we really liked this work. The authors should be congratulated for a thorough line of thoughtful experiments in support of their regulatory model as mentioned above. The manuscript is clearly written and we enjoyed reading it. We have few general comments/suggestions that should be addressed/clarified.

1) Are the changes in *Rnpc3* observed in tissues in the same range as done in the titration experiments?

2) Expression computation and correlations: It's not clear how these were computed and whether these were done properly. The authors state TPM were derived by Whippet (Figure 3—figure supplement 1) but Whippet is designed only for splicing changes. It's not clear how Whippet would give full transcripts, and more importantly weighted gene level TPM values. Furthermore, TPM is not a proper measure to compare across many different experiments/conditions (it's not as bad as RPKM but still not great). Between sample normalization should be applied as implemented in DESeq and TMM. See for example https://haroldpimentel.wordpress.com/2014/12/08/in-rna-seq-2-2-between-sample-normalization/

3) Figure 4: We understand where these p-values come from, but we are still worried about possible artifacts in the normalization procedures that might affect the results (also see above). Another way to compute a p-value and address the above concern is to compute an empirical p-value compared to sampling a large set of similarly expressed genes and computing the correlation values for them. True, some may be bona fide targets as well, but presumably this population of targets is rather small and the Sr proteins correlations stand out.

---

## [Author Response]

Reviewer #2:This paper describes an unusual regulatory feedback process in the alternative splicing of the Srsf10 gene involving competition between major and minor intron splice sites. Several RNA binding proteins self-regulate their protein levels by inclusion of alternative "poison exons" that include premature stop codons. Srsf10 appears to have a poison exon 3 who's exclusion requires minor spliceosome activity. This part of the paper seems solid. They then show that there is a general correlation of the expression levels of many SR proteins including Srsf10. This has been described before and similar correlations can be seen in ribosomal protein genes indicating that they are under coordinate regulation. The authors then remove the poison exon from the Srsf10 gene and argue that the results support a key role for Srsf10 and, by extension, the minor spliceosome in controlling the mRNA levels of SR genes in general. The key results shown in Figure 4E do not, to me, support this idea. The authors speculate that alterations in mRNA levels are due to NMD. A direct demonstration would be more convincing. With the data they show, I think they should focus on the Srsf10 case alone rather than pushing for a more global mechanism.

We agree that the effects observed in Figure 4F are rather subtle. However, we find it quite remarkable that the expression of all SR proteins is increased in the Crispr cell line and for 8/12 this is statistically significant. We would therefore like to keep this figure, but have modified the text to say “…was sufficient for subtle upregulation…”. We have also weakened the conclusion, which now reads: “…which is sufficient to change expression levels of the other SR proteins in a tissue- and differentiation state-specific manner.” In the previous version it was “of all other SR proteins”. We hope that these adjustments address the reviewer’s concern.

Reviewer #3:[…]Overall, we really liked this work. The authors should be congratulated for a thorough line of thoughtful experiments in support of their regulatory model as mentioned above. The manuscript is clearly written and we enjoyed reading it. We have few general comments/suggestions that should be addressed/clarified.1) Are the changes in Rnpc3 observed in tissues in the same range as done in the titration experiments?

In cell culture siRNA mediated knockdown of *Rnpc3* results in reduction of the *Rnpc3* mRNA levels to 50%. Under this conditions *Srsf10* levels are reduced to ~75%. Using the slope of the linear regression fit for *Rnpc3* and *Srsf10* across different tissues we can calculate how strong a 2-fold reduction in *Rnpc3* effects *Srsf10*. Based on this calculation *Srsf10* expression is reduced to 55%, which is in good agreement with the 75% observed in cell culture.

2) Expression computation and correlations: It's not clear how these were computed and whether these were done properly. The authors state TPM were derived by Whippet (Figure 3—figure supplement 1) but Whippet is designed only for splicing changes. It's not clear how Whippet would give full transcripts, and more importantly weighted gene level TPM values. Furthermore, TPM is not a proper measure to compare across many different experiments/conditions (It's not as bad as RPKM but still not great). Between sample normalization should be applied as implemented in DESeq and TMM. See for example https://haroldpimentel.wordpress.com/2014/12/08/in-rna-seq-2-2-between-sample-normalization/

Thank you for raising these points, which are important for our conclusions and thus merit further clarification. First, the Whippet Quant function also calculates weighted TPM values on gene level (see Sterne-Weiler, 2018 and the GitHub documentation). This is a great advantage of the Whippet tool, as it allows alignment, splicing, and gene expression analysis in one step/one pipeline.

However, to be more confident with our results we repeated the analyses using Salmon to determine GE levels and DESeq2 to obtain normalized expression counts. These analyses were performed by an additional bioinformatician, Alexander Neumann, who is now included as co-author. Comparing expression across different tissues, we observe comparable correlations using Salmon derived gene expression levels, which are now included as Figure 3—figure supplement 1B (*Srsf10*) and 1C (*Gapdh*). When removing a single outlier value, we also observe a significant correlation between *Rnpc3* and *Srsf10* in the DESeq2 normalized values (R² = 0.7509, p < 0.0001) but as we observe large variations (e.g. the three *Rnpc3* values in epididymis 1: 222; 2: 17921; 3: 2655 and *Srsf10* 1: 2522; 2: 28859; 3: 6516) we would rather not include these into our manuscript. This is also mentioned in the Materials and methods part: “DESeq2 based normalized expression counts dramatically increased the variance of replicate tissue samples (but not of ES cell differentiation), which is why we chose to analyze only the un-normalized values for the tissue comparisons.”

In ES cell differentiation, we also observe comparable correlations using Salmon derived gene expression levels (new Figure 3—figure supplement 1I). Here however, normalization via DESeq2 strongly increases the correlation (new Figure 3—figure supplement 1J), making us confident that the DeSeq2 analysis in principle works (but is not suitable for the different tissue samples). Taken together, these additional analyses are fully consistent with the conclusion that *Rnpc3* and *Srsf10* expression indeed correlate across different tissues and development stages.

3) Figure 4: We understand where these p-values come from, but we are still worried about possible artifacts in the normalization procedures that might affect the results (also see above). Another way to compute a p-value and address the above concern is to compute an empirical p-value compared to sampling a large set of similarly expressed genes and computing the correlation values for them. True, some may be bona fide targets as well, but presumably this population of targets is rather small and the Sr proteins correlations stand out.

The R² and p-values for correlations of SR-protein and RNPC3 expression across different tissues indeed stands out with most R² values above 0.6 (average 0.74) and each p-value < 0.0001. These correlations are slightly reduced with Salmon derived correlations (R² average 0.6) but remain always highly significant (each p-value < 0.0001). Additionally, we would like to point out that in Figure 3D we have calculated correlation coefficients for 587 minor intron containing genes and 629 only major intron containing genes using the same procedure. This resulted in mean R² values of 0.24 and 0.16, respectively. We are therefore confident that the correlation coefficients for SR proteins are indeed true, as they stand out from only major intron containing genes and also from other minor intron containing genes. Also in ES cell differentiation a mean R² of 0.37 for SR proteins is clearly higher than the correlation for all major intron genes (0.16).